# Structure, Activity, and Function of SETMAR Protein Lysine Methyltransferase

**DOI:** 10.3390/life11121342

**Published:** 2021-12-04

**Authors:** Michael Tellier

**Affiliations:** Sir William Dunn School of Pathology, University of Oxford, South Parks Road, Oxford OX1 3RE, UK; michael.tellier@path.ox.ac.uk; Tel.: +44-1865-275583

**Keywords:** SETMAR, Metnase, H3K36me2, Hsmar1, non-homologous end joining repair, NHEJ, transposase, transposable elements, histone, methyltransferase

## Abstract

SETMAR is a protein lysine methyltransferase that is involved in several DNA processes, including DNA repair via the non-homologous end joining (NHEJ) pathway, regulation of gene expression, illegitimate DNA integration, and DNA decatenation. However, SETMAR is an atypical protein lysine methyltransferase since in anthropoid primates, the SET domain is fused to an inactive DNA transposase. The presence of the DNA transposase domain confers to SETMAR a DNA binding activity towards the remnants of its transposable element, which has resulted in the emergence of a gene regulatory function. Both the SET and the DNA transposase domains are involved in the different cellular roles of SETMAR, indicating the presence of novel and specific functions in anthropoid primates. In addition, SETMAR is dysregulated in different types of cancer, indicating a potential pathological role. While some light has been shed on SETMAR functions, more research and new tools are needed to better understand the cellular activities of SETMAR and to investigate the therapeutic potential of SETMAR.

## 1. Introduction

In eukaryotes, DNA is wrapped around proteins called histone to form the chromatin, a nucleoprotein structure. The basic unit of chromatin is the nucleosome, which is composed of 147 base pairs (bp) and eight histone proteins: two histones H2A, two H2B, two H3, and two H4 [1]. An additional histone, H1, is positioned between nucleosomes to regulate their packaging. DNA processes, such as transcription, replication, or DNA repair, are therefore dependent on chromatin remodeling [2]. A key factor in chromatin regulation is the addition, removal, and reading of post-translational modifications (PTMs) that can be deposited on the tails of histones, especially H3 and H4. Frequent histone PTMs include methylation, acetylation, phosphorylation, or ubiquitination but numerous other PTMs have also been described, referred as a whole as the histone code [3]. The histone code hypothesis implies that specific patterns of histone PTMs are associated and involved in specific DNA processes. For example, euchromatin, also known as open chromatin, is associated with active transcription and histone marks such as histone H3 lysine 4 trimethylation (H3K4me3), H3K36me3, H3K79 mono- di-, and trimethylation (me1, me2, me3), and acetylation of H3K9 or H3K27 (H3K9ac or H3K27ac). In contrast, heterochromatin, or close chromatin, is linked to inactive transcription and the histone marks H3K9me3 and H3K27me3 [4,5].

Dimethylation of H3K36 (H3K36me2) is associated with regulation of gene expression and DNA damage repair [6]. In humans, H3K36me2 is catalyzed by several histone methyltransferases, including ASH1L, NSD1-3, SETD3, SETMAR, and SMYD2 [7]. The interest in SETMAR has started to grow following the sequencing of the human genome as *SETMAR* was found to be one of the 47 genes containing or derived from a domesticated transposable element [8]. Later works have shown that in mammals, SETMAR exists in two different versions, as a single SET histone methyltransferase domain in most mammals while in anthropoid primates, SETMAR is a fusion between the SET domain and an inactive domesticated DNA transposase, Hsmar1 [8,9,10,11]. SETMAR is expressed ubiquitously and has been found to dimethylate H3K4 and H3K36 [12]. Since its discovery, the human SETMAR has been associated with numerous cellular processes, including DNA damage repair via the non-homologous end joining (NHEJ) pathway, illegitimate DNA integration, restart of stalled replication forks, chromosomal decatenation, suppression of chromosomal translocations, and regulation of gene expression [12,13,14,15,16,17,18,19,20,21]. In addition, SETMAR is dysregulated in several cancers, such as glioblastoma, leukemia, hematologic neoplasms, breast and colon cancer, and mantle cell lymphoma [16,22,23,24,25,26,27,28,29,30]. 

## 2. Structural Features

### 2.1. Domain Architecture

SETMAR is composed of two domains, the protein lysine methyltransferase (SET) domain and the DNA transposase (MAR) domain (Figure 1A). The protein lysine methyltransferase domain is constituted by the SET subdomain, named after the Su(var)3–9, enhancer-of-zeste and Trithorax (SET) proteins originally identified in Drosophila [31,32,33]. Two other subdomains, the pre-SET and the post-SET flank the SET subdomain. The SET domain contains binding sites for the lysine ligand and the co-factor S-adenosylmethionine (SAM), which provides the methyl groups.

The DNA transposase domain of SETMAR is catalytically inactive, as several inactivating mutations in the catalytic subdomain have neutralized the DNA transposition activity of SETMAR [10,11,34]. In contrast, the DNA binding subdomain is under strong purifying selection [9] and retains its binding activity to the inverted-terminal repeats (ITRs) of Hsmar1 transposon remnants both in vitro [9] and in SETMAR chromatin immunoprecipitation followed by sequencing (ChIP-seq) experiments [21,35,36]. The DNA binding specificity of SETMAR is regulated via an interaction with PRPF19, with the SETMAR-PRPF19 complex able to bind non-ITRs sequences [37]. Similarly to the ancestral Hsmar1 DNA transposase, the DNA transposase domain of SETMAR also forms a homodimer [38]. While SETMAR cannot perform DNA transposition, it was shown to perform in vitro 5′end nicking on Hsmar1 transposon ends, in presence of DMSO and Mn^2+^ [10], and to act as an endonuclease, such as Artemis, trimming DNA overhangs [39,40,41]. 

### 2.2. Isoforms

The *SETMAR* gene encodes eight different mRNA isoforms in human tissues with variable combinations of domains and subdomains (Figure 1B). Surprisingly, only one isoform encodes for a SETMAR protein containing both the SET domain and the DNA transposase domain (ENST00000358065.4). The remaining isoforms encode proteins containing only the DNA transposase domain (three isoforms), only an active SET domain (one isoform), or no complete domain (three isoforms). Out of the eight isoforms, four isoforms, encoding full-length SETMAR, SET domain only, or the DNA transposase domain only, are dominantly expressed in tissues. Interestingly, the most expressed isoform encodes a SETMAR protein containing only the DNA transposase domain. Two splicing factors have been found to regulate SETMAR alternative splicing, NONO and SFPQ, in bladder cancer [30]. As these two splicing factors are ubiquitously expressed, NONO and SFPQ are likely to be general regulators of SETMAR alternative splicing. 

### 2.3. Structure

The complete structure of SETMAR has not been determined yet. Currently, only structures of the SET domain and of the DNA transposase catalytic subdomain have been resolved (Figure 2A,B) [38,42]. In agreement with other eukaryotic DNA transposases, such as Mos1 [43], the MAR domain of SETMAR forms a homodimer (Figure 2B). A potential model of a SETMAR full-length isoform is provided by AlphaFold (Figure 2C) [44]. However, the model represents only a monomer while SETMAR is known to form a dimer via the DNA transposase domain. The structures of full-length SETMAR homodimer and of combination between the different isoforms (full-length/SET-depleted isoform) remain to be obtained to understand better how SETMAR methyltransferase activity is regulated.

## 3. Biological Roles of SETMAR

### 3.1. Substrates

In addition to its own automethylation [15,45], SETMAR has been found to methylate snRNP70, a pre-mRNA splicing factor, and SPTBN2, a spectrin [45], and to catalyze H3K4me2 [12], H3K27me3 [30], and H3K36me2 [12,19,21,28]. 

The H3K36me2 histone methyltransferase activity of SETMAR has been the most intensively investigated. The initial study found that SETMAR could dimethylate H3K36, and H3K4 to a lesser extent, in vitro [12]. In a later study investigating the link between H3K36me2 and NHEJ in HT1904 cells, the authors have shown that overexpression or knockdown of SETMAR is associated with an increased or decreased H3K36me2 level by ChIP around DNA double strand breaks (DSBs), respectively [19]. Furthermore, overexpression of a methyltransferase-deficient SETMAR, D261S, compared to WT SETMAR fails to induce H3K36me2 around DSBs [19]. However, another study could not replicate SETMAR dimethylation of H3K36 in vitro by LC-MS/MS proteomics and Western blot [45]. Carlson et al. found that SETMAR could only weakly methylate histone H3 in vitro, potentially on H3K115, a residue located in the DNA-histone dyad interface [45,46]. A study in U2OS cells revealed that while overexpression of WT SETMAR did not increase H3K36me2 level, overexpression of a methyltransferase-deficient SETMAR, N223A, was associated with a decreased H3K36me2 level by Western blot and ChIP [21]. Another study in glioblastoma cell lines reported that knockdown of SETMAR by shRNA or siRNA is associated with a decreased H3K36me2 level [28]. A recent study in bladder cancer found that SETMAR could also mediate H3K27me3 [30], a mark associated with heterochromatin and gene repression. 

As non-histone targets, SETMAR was found to perform automethylation on two different residues, lysine 335 and lysine 498 [15,45]. However, each residue has been found to be methylated in only one study. Lysine 335 methylation did not affect SETMAR methyltransferase activity but might be involved in the regulation of protein-protein interactions [45]. In contrast, methylation of lysine 498 was shown to inhibit SETMAR activity in chromosome decatenation [15].Carlson et al. found that SETMAR could methylate two other non-histone targets, the splicing factor snRNP70 on the lysine 170 and SPTBN2, a spectrin (residue not provided) [45]. However, it remains unknown whether the SETMAR mediated methylation affects the activities of these proteins.

### 3.2. Regulation

The expression of SETMAR has been found to be regulated by SOX11 [23], a transcription factor involved in development, including neurogenesis and skeletogenesis, and disease, such as neurodevelopmental disorders, osteoarthiritis, and cancers [47]. The Notch signaling pathway, involved in developmental and homeostatic processes [48], also regulates SETMAR expression as knockdown of each of the four Notch receptors decreases SETMAR expression in colon cancer stem cells [24]. 

SETMAR activities have been proposed to be regulated by different mechanisms including automethylation, phopshorylation, the presence of an autoinhibitory loop from the post-SET domain, and by alternative splicing. Automethylation of SETMAR has been observed on two residues, lysine 335 and lysine 498 [15,45]. While only lysine 498 was found to regulate SETMAR activity in DNA decatenation, the mechanism behind this automethylation regulation remains unclear. Following DNA damage, SETMAR is phosphorylated on its Ser508 residue, located in the catalytic subdomain of the transposase domain, by Chk1 and is dephosphorylated by PP2A [49]. Overexpression of a Ser508 to alanine mutant compared to overexpression of wild-type SETMAR results in a decreased association of SETMAR to DSBs, a reduced DSB repair in vivo, and a higher nuclease activity in vitro [49]. Of note, phosphorylation of the equivalent residue in another transposase, Ser170 in Mos1, is performed by the cAMP-dependent protein kinase (PKA) and prevents the active transport of the transposase to the nucleus and also interferes with the formation of the paired-end complex (a transposase dimer bound to two ITRs) [50]. SETMAR binding to ITRs might therefore be also regulated by phosphorylation on its Ser508 residue. 

Another regulation of SETMAR histone methyltransferase activity is provided by the loop connecting the SET and post-SET domains that can take an autoinhibitory conformation that will interfere with the accession of the histone tail substrate. The presence of an autoinhibitory loop is not specific to SETMAR as it has also been found in other H3K36 methyltransferases, such as NSD1, SETD2, or ASH1L [51,52,53,54]. Interaction with a nucleosome is thought to stabilize an active conformation of the post-SET loop, promoting methylation of H3K36 [51]. However, it remains to be determined whether this autoinhibitory loop also regulates the methylation of non-histone targets.

SETMAR is also regulated by alternative splicing, which can produce methyltransferase deficient isoforms without a complete SET domain (Figure 1B). Two splicing factors were found to regulate SETMAR alternative splicing, NONO and SFPQ, with a knockdown of NONO promoting the production of the methyltransferase deficient SETMAR isoform over the full-length isoform [30]. In addition, the use of an alternative TSS, which adds 13 amino acid, has been associated with an increased protein stability [27]. 

### 3.3. Sequence Specificity

The sequence specificity of SETMAR methyltransferase activity remains unclear due to the low number of substrates that have been found. However, based on the methylation proteomics performed by Carlson et al., it has been proposed that SETMAR could recognize the following consensus sequence on proteins: KR(I/L) [45].

### 3.4. Connection to Cell Signaling Pathways

Excluding the regulation of SETMAR expression by the Notch signaling pathway in colon cancer stem cells [24], SETMAR has not been directly connected to any other specific cell signaling pathway. However, SETMAR is phosphorylated on the residue Ser508 by Chk1, a serine/threonine kinase that coordinates DNA damage response and cell cycle checkpoint response [55]. Phosphorylation of the Ser508 residue is associated with a better recruitment of SETMAR to DSBs and an increased DNA repair [49]. In turn, phosphorylated SETMAR increases Chk1 stability as it interferes with the interaction between DDB1 and Chk1, which is required for ubiquitination of Chk1 and its degradation via the proteasome [56]. In addition, overexpression of the wild-type full-length SETMAR in the U2OS cell line resulted in transcriptional changes that are enriched for five signaling pathways: Rap1, PI3K-Akt, calcium, cAMP, and Hippo [21]. Further work is required but SETMAR might be involved in the regulation of some cellular signaling pathways. 

### 3.5. Connection to Chromatin Regulation

Dimethylation of H3K36 is associated with several chromatin processes, including regulation of gene expression through controlling the distribution of H3K27me3 and DNA methylation [57], and DNA damage repair [6]. SETMAR has been associated with both regulation of gene expression and repair of DSB by the NHEJ pathway [19,21]. A recent paper has associated expression of the full-length SETMAR, which contains the active SET domain, with an increased H3K27me3 level [30]. However, it remains unclear whether SETMAR could trimethylate H3K27 or the change in H3K27me3 level is a consequence of transcriptional change due to the expression of the full-length SETMAR, as H3K36me2/me3 and H3K27me3 are known to be anti-correlated [58]. In addition, H3K36me2 has been shown to recruit the DNA methyltransferase DNMT3A to maintain DNA methylation at non-coding regions of euchromatin [57]. It will, therefore, be of interest to investigate whether SETMAR mediated H3K36me2 could also regulate DNA methylation maintenance, especially around Hsmar1 ITRs bound by SETMAR.

### 3.6. Cellular Roles and Function

In mouse, where SETMAR contains only the SET domain, homozygous deletion is associated with several phenotypic defects in vision/eye, behavior/neurological phenotype, metabolism, pigmentation, skeleton phenotype, and immune system, indicating a developmental role for the SET domain [59]. In humans, SETMAR is involved in several cellular functions, including DNA damage repair via the NHEJ, DNA integration, cell cycle and DNA replication, and the regulation of gene expression [12,13,14,15,16,17,18,19,20,21]. SETMAR activities are regulated via protein-protein interactions, such as the splicing factors PRPF19 and SPF27 [37], the DNA damage repair factors PRPF19, DNA ligase IV, and XRCC4 [13], and the DNA replication factors topoisomerase 2α (TOP2A), PCNA, and RAD9 [17]. 

The NHEJ pathway is one of the four pathways used by a cell to resolve DSBs but, in comparison to homologous recombination, can result in mutations and insertions-deletions [60]. Illegitimate DNA integration, such as genomic insertion of a plasmid, is also mediated by the NHEJ [61]. SETMAR has been proposed to act in most steps of NHEJ (Figure 3). Following DNA damage, SETMAR is phosphorylated on Ser508 by Chk1 [49] and recruited by PRPF19 to DSBs [37] where it dimethylates H3K36 on the surrounding nucleosomes [19]. Increased level of H3K36me2 helps to recruit and stabilize Ku70 and NBS1 to the DNA free ends, promoting DNA repair [19]. In addition to the dimethylation of H3K36, the catalytic subdomain of the transposase domain of SETMAR has also been proposed to act as an endonuclease, such as Artemis, to trim DNA overhangs [39,40,62]. Processing of non-compatible ends is required before ligation by the DNA ligase IV and XRCC4 complex can happen. As SETMAR can interact with DNA ligase IV and XRCC4 [13], it is thought that SETMAR could also help to recruit the ligation complex. A recent work with an in vivo system to follow NHEJ repair and cell lines overexpressing either SETMAR or its domains alone found that, while each SETMAR domain has an effect on NHEJ, no change was observed with the full-length SETMAR [20]. More works with better in vivo approaches are still required to properly understand the roles of SETMAR in NHEJ (see Part 5). 

SETMAR has also been proposed to act in DNA replication and cell cycle regulation (Figure 4A). Knockdown, knockout, or overexpression of SETMAR has more or less of an effect on cell cycle depending on the cell type, with generally a higher SETMAR expression correlating with an increased growth rate [17,20,22,45,63]. While phosphorylation of SETMAR on the Ser508 residue by Chk1 is associated with DNA damage repair via NHEJ, the unphosphorylated form of SETMAR is involved in the response to replication stress [49]. The involvement of SETMAR in replication fork restart has been shown via interactions between SETMAR and PCNA and RAD9, and by a higher sensitivity of cells, following knockdown of SETMAR, to hydroxyurea (HU), a treatment that decreases the production of nucleotides and therefore induces replication stress [17,41,64,65]. In contrast, a recent CRISPR/Cas9 screening against DNA-damaging agents did not find SETMAR as a gene increasing the sensitivity of cells to HU [66]. The endonuclease activity of SETMAR is thought to be involved in the cleavage of branched DNA structures resulting from stress replication forks, producing DSBs that can be resolved by DNA damage repair pathways and thus allowing the restart of replication forks [17,41,64,65]. A recent work has proposed that SETMAR role in the response to replication stress was mediated by the dimethylation of H3K36 at stalled replication forks, facilitating the recruitment of DNA repair factors, rather than a cleavage of stalled forks via SETMAR endonuclease activity [63]. It has been recently suggested that SETMAR could bind to 12 bp motifs that are enriched in the regions of replication origins but more work is required to determine whether SETMAR binding is occurring simultaneously to DNA replication and whether SETMAR binding plays a role in DNA replication [35]. In addition, SETMAR interaction with TOP2A has been found to enhance TOP2A function in chromosome decatenation and to promote resistance to topoisomerase II inhibitors in breast cancer cells (Figure 4B) [15,16]. SETMAR mediated enhancement of TOP2A function in chromosome decatenation is negatively regulated by the automethylation of SETMAR on its residue lysine 498 [15].

Cordaux et al. proposed in one of the first papers published on SETMAR that the remnants of the Hsmar1 transposable element, including SETMAR and the Hsmar1 binding sites scattered across the genome, could represent a case of a new gene regulatory network with SETMAR binding Hsmar1 ITRs to regulate gene expression via histone methylation [9,67]. This hypothesis was confirmed later with three papers showing that SETMAR binds Hsmar1 ITRs in vivo [21,35,36], and that the overexpression of full-length SETMAR in U2OS cells upregulated the expression of 960 genes, that are enriched with genes containing Hsmar1 binding sites, while overexpression of a methyltransferase dead SETMAR hardly affected gene expression [21]. However, the mechanism behind SETMAR function in gene expression remains unclear as no consistent changes in H3K36me2 level was observed across upregulated genes [21] and SETMAR does not generally bind to promoter regions [21,35,36]. Regulation of gene expression could be mediated via binding of SETMAR to enhancers (Figure 5A), which are non-coding RNA genes regulating promoter activity via enhancer-promoter loops, and/or via regulation of the 3D genome organization (Figure 5B), as SETMAR could bind two Hsmar1 remnants located apart on the same chromosome or on different chromosomes. It has been recently shown that some enhancers contain SETMAR binding sites [35] but more work is required to determine whether SETMAR binding could affect the activity of these enhancers and the expression of their associated protein-coding genes. 

## 4. Connection to Diseases

While some mutations have been found in *SETMAR* in different cancers (Figure 6A), no recurrent mutations have been observed. A recent study on 100 colon cancer samples with high microsatellite instability found only one sample with a single frameshift mutation, c.1409delA, in *SETMAR* [29]. Of note is the C226S mutation (Figure 6A) that has been found in five ovarian cancer samples as the mutation is located within the conserved NHSC motif of the SET domain. In cell lines, the mutation N223A, located in the same motif, decreases H3K36me2 level and affects gene expression and DNA repair [12,20,21], which might be mimicked by the C226S mutation. 

Dysregulation of SETMAR expression has been associated with several cancers, including glioblastoma, leukemia, hematologic neoplasms, breast and colon cancer, and mantle cell lymphoma (Figure 6B) [16,22,23,24,25,26,27,28,29,30]. Overexpression of the full-length wild-type SETMAR in the osteosarcoma U2OS cell line is associated with broad changes in the transcriptome that are enriched in pathways connected to cancer, such as angiogenesis or response to hypoxia, and signaling pathways linked to cellular proliferation, such as the PI3K-AKT and the Hippo pathways [21,69]. In bladder cancer cells, expression of the full-length SETMAR has been associated with an inhibition of lymph node metastasis via an increase in H3K27me3 at the promoters of metastatic oncogenes, inhibiting their transcription [30]. In addition, SETMAR, via its role in the NHEJ pathway, is important in the survival of glioblastoma cancer cells to radiation therapy, which can cause cancer relapse [28]. 

On the treatment side, knockdown of SETMAR increases the sensitivity of leukemia cell lines to etoposide [22] and of breast cancer cell lines to the anthracycline Adriamycin [16]. Ciprofloxacin, a Quinolone drug acting on bacterial DNA gyrase, inhibits the NHEJ activity of the transposase domain of SETMAR [70]. Combination of ciprofloxacin and cisplatin shows a higher efficacy against the A549 cancer cell line, both in tissue culture and in a mouse A549 xenograft model, compared to ciprofloxacin or cisplatin alone [70]. However, the cytotoxic activity of ciprofloxacin against cancerous cells requires a high, non-pharmacological concentration, which currently restricts its use as a potential cancer treatment [71,72,73].

## 5. Directions for Future Research

After ~15 years of investigation on SETMAR functions, many questions remain unanswered. While specific antibodies against SETMAR are available [27,74], a major limitation remains the current lack of tools to study SETMAR in vivo. Current in vivo works have used knockdown, knockout, and overexpression approaches, which limit the interpretation of the primary functions of SETMAR as secondary/indirect effects are present. It will, therefore, be important in future studies to use, for example, endogenous targeted degradation approaches, such as auxin-inducible degradation [75] or the dTAG system [76], to obtain a quick and specific degradation of SETMAR. In addition, these targeted degradation systems could also help to investigate the functions of SETMAR isoforms that are expressed from alternative TSSs and poly(A) sites. 

Current efforts in targeting SETMAR has found ciprofloxacin as an inhibitor of SETMAR transposase domain [70], but the cytotoxicity of this small molecule remains an issue [71,72,73]. The development of less cytotoxic small molecules targeting SETMAR DNA transposase domain is still needed before potential clinical use. Another important need will be the development of a specific inhibitor of SETMAR methyltransferase activity, which will be critical to determine the targets of SETMAR. In addition, a specific inhibitor could potentially be clinically relevant, by itself or in combination, as SETMAR is dysregulated in several cancers. Another possibility could be the development of small molecules promoting the ubiquitination and degradation of SETMAR by the proteasome, such as PROTACs or molecular glues [77]. 

As SETMAR is known to form a dimer and that out of the four most expressed isoforms, three are able to dimerize, a better understanding of the isoforms combination that are present in SETMAR dimers is required. Another significant question is why two out of the four expressed isoforms encodes only for the DNA transposase domain? With the exception of the viral vSET [78], human histone methyltransferases exists usually as monomers [79]. However, the DNA transposase domain of SETMAR enforces the formation of a dimer, which could result in the presence of two SET domains, potentially affecting the methyltransferase activity because of steric clashes. Interestingly, the phylogenetic analysis of a SETMAR isoform encoding a deleted SET domain and an active DNA binding domain (ENST00000413809.5) shows that this isoform is specific to anthropoid primates that contains the domesticated DNA transposase domain [20]. Therefore, the presence of multiple SETMAR isoforms encoding for a deleted SET domain could represent a way to decrease the possibility of a SETMAR dimer to contain two SET domains. However, a global phylogenetic analysis of the different SETMAR isoforms across mammals is required to determine whether all the SET deleted isoforms are specific to the species that contain the domesticated DNA Hsmar1 transposase. In addition, it will be important to determine which SETMAR dimers are active, i.e., SETMAR dimers with two SET domains, with only one SET domain, or without the SET domain.

Another question is whether SETMAR could be involved in pre-mRNA splicing and/or alternative splicing. In addition to methylating snRNP70, which is part of the U1 snRNP that recognizes the 5′ splice site [45], SETMAR also interacts with two pre-mRNA splicing factors, PRPF19 and SPF27, that are part of the NineTeen Complex (NPC) [37]. While overexpression of wild-type SETMAR or of a methyltransferase-deficient SETMAR does not affect the exon inclusion/exclusion near SETMAR binding sites [21], it remains to be determined if the methylation of snRNP70 or the modulation of SETMAR expression level or methyltransferase activity affect pre-mRNA splicing.

More research is also needed to determine whether SETMAR could be involved in development. In mice, SETMAR homozygous knockout results in developmental defects [59]. In humans cells, SETMAR expression is regulated by SOX11 [23], a transcription factor involved in neurogenesis and skeletogenesis [47], while overexpression of SETMAR in U2OS cells results in transcriptional changes in numerous genes involved in development, especially brain development [21]. 

## Figures and Tables

**Figure 1 life-11-01342-f001:**
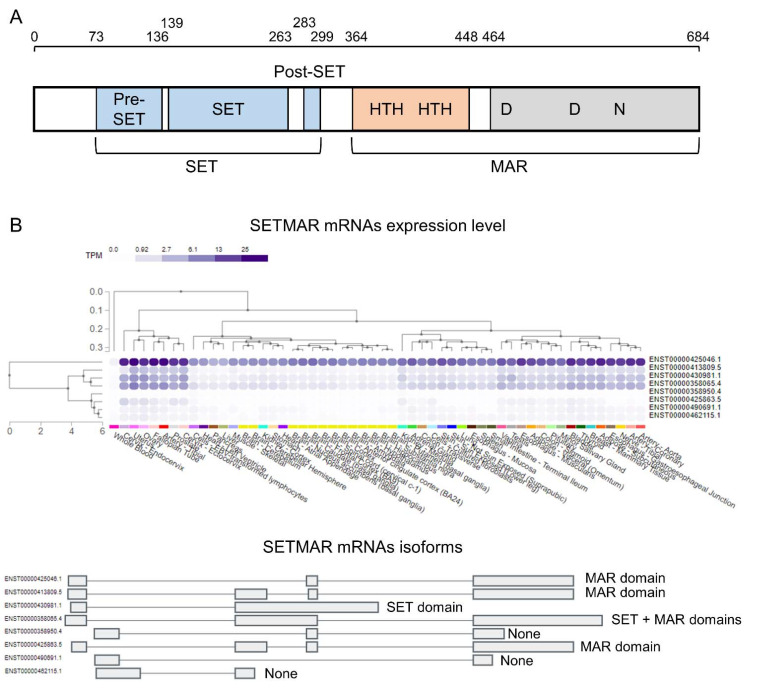
Protein organization and expression level of SETMAR in human. (**A**) SETMAR is composed of two domains, the protein methyltransferase domain (SET) that includes the pre-SET, the SET- and the post-SET subdomains, and the DNA transposase domain (MAR) that contains the DNA binding (orange) and the catalytic (gray) subdomains. (**B**) *SETMAR* encodes for eight different isoforms. One isoform encodes for the full-length SETMAR, one isoform includes only the SET domain, and three isoforms contain only the DNA transposase domain. Out of the eight mRNA isoforms, only four are expressed at a high level in human tissues (from GTEx).

**Figure 2 life-11-01342-f002:**
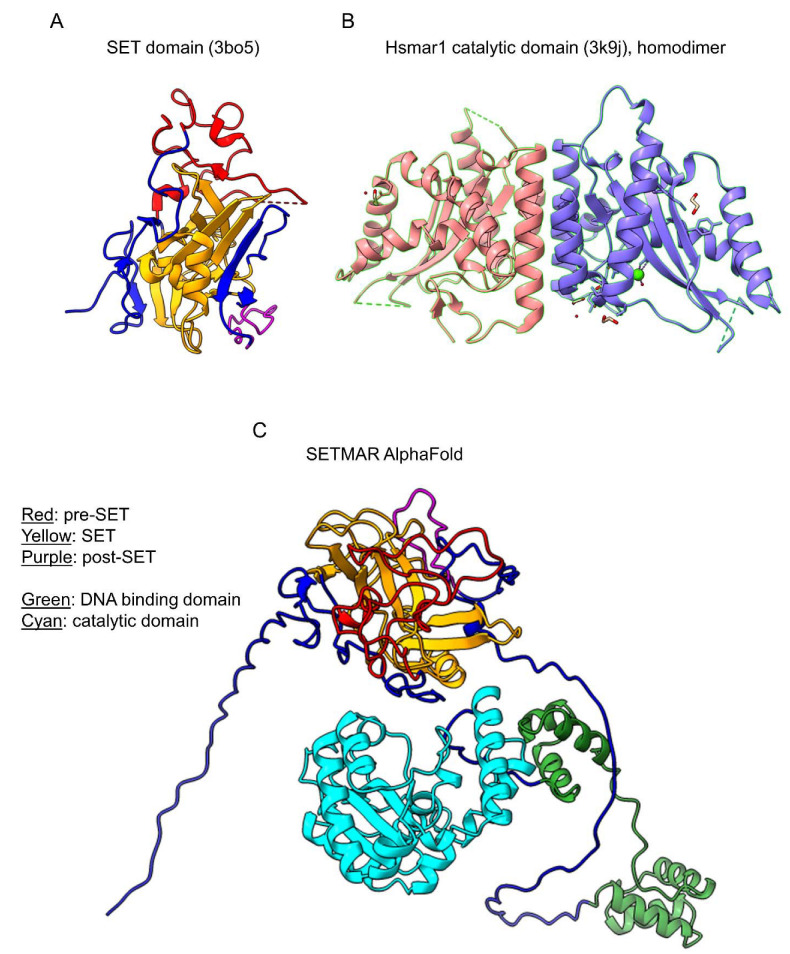
Crystal structures of SETMAR. (**A**) Structure of the SET domain of SETMAR with the pre-SET, SET, and post-SET subdomains shown in red, yellow, and purple, respectively. (**B**) Structure of the catalytic subdomain of the DNA transposase domain of SETMAR, shown as a homodimer. Each monomer is represented by a different color. (**B**,**C**) Proposed structure of a SETMAR monomer by AlphaFold with the pre-SET, SET, and post-SET subdomains shown in red, yellow, and purple, respectively, and the DNA binding and catalytic subdomains shown in green and cyan, respectively.

**Figure 3 life-11-01342-f003:**
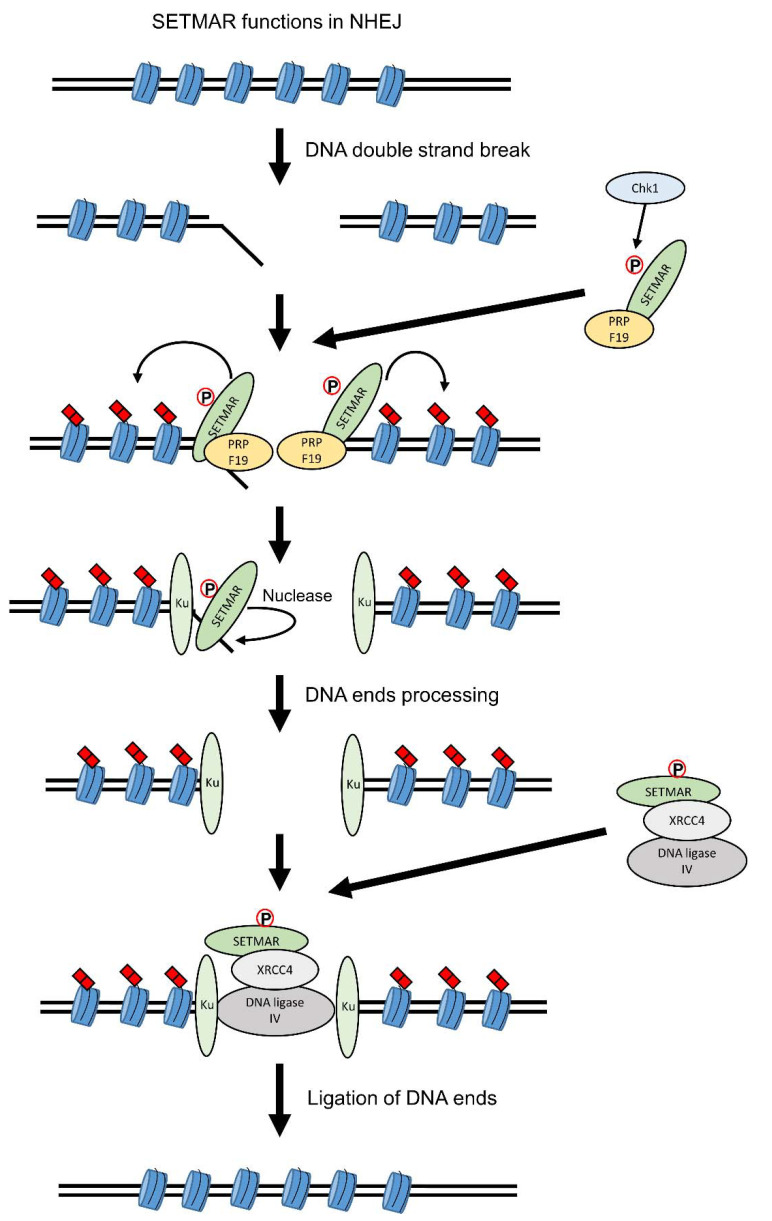
SETMAR functions in non-homologous end joining (NHEJ). SETMAR has been found to act on several aspects of the NHEJ pathway. Following DNA double-strand breaks (DSBs), SETMAR is phosphorylated by Chk1 on its Ser508 residue and interacts with PRPF19. The SETMAR-PRPF19 complex binds to the DSB site and SETMAR dimethylates H3K36 on the neighboring nucleosomes. H3K36me2 stabilizes the association of Ku70 and NBS1 to DSB sites. SETMAR is also involved in the processing of the DNA ends via its endonuclease activity, similarly to Artemis. Following DNA ends processing, SETMAR is also involved in the ligation of the two DNA ends as it interacts with the DNA ligase IV and XRCC4 complex. For the readability of the figure, the proteins are shown as monomer.

**Figure 4 life-11-01342-f004:**
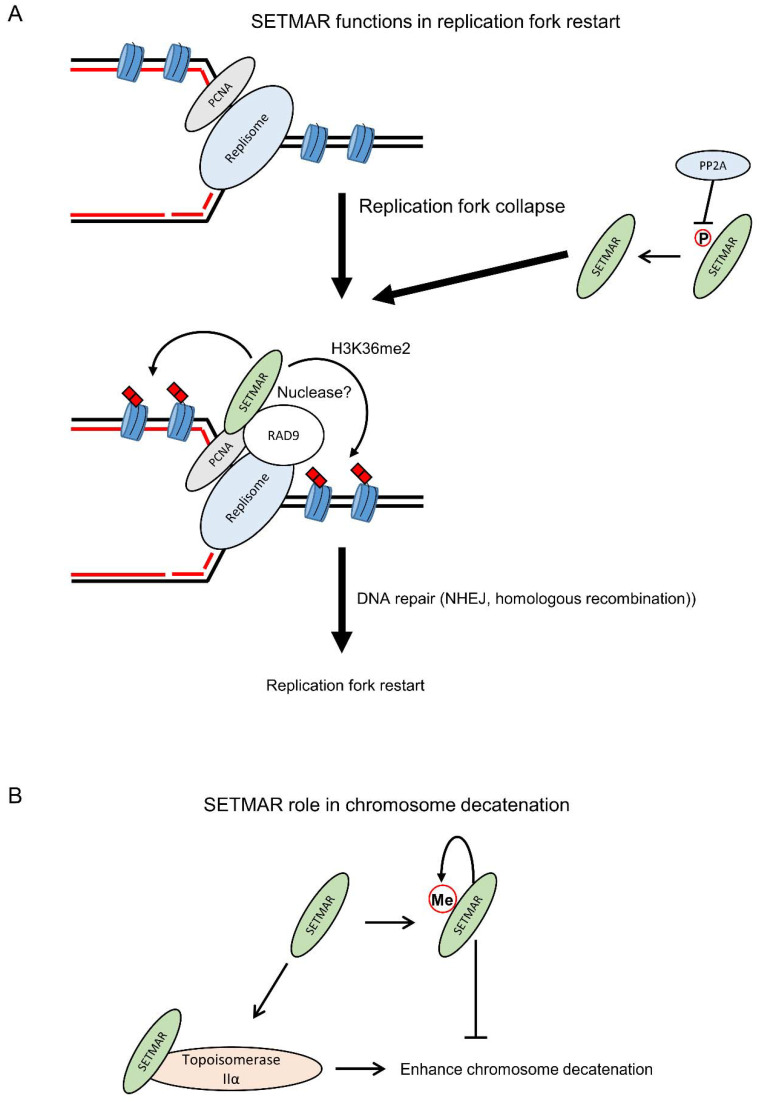
SETMAR functions in replication fork restart and chromosome decatenation. (**A**) Following replication fork collapse, the unphosphorylated form of SETMAR, mediated by the protein phosphatase PP2A, can interact with PCNA and RAD9 to promote replication fork restart via the dimethylation of H3K36, which helps recruiting DNA repair factors, and its endonuclease activity. (**B**) SETMAR can also enhance chromosome decatenation via its interaction with TOP2A. SETMAR activity in chromosome decatenation is regulated through its automethylation on the lysine 498 residue. For the readability of the figure, the proteins are shown as monomer.

**Figure 5 life-11-01342-f005:**
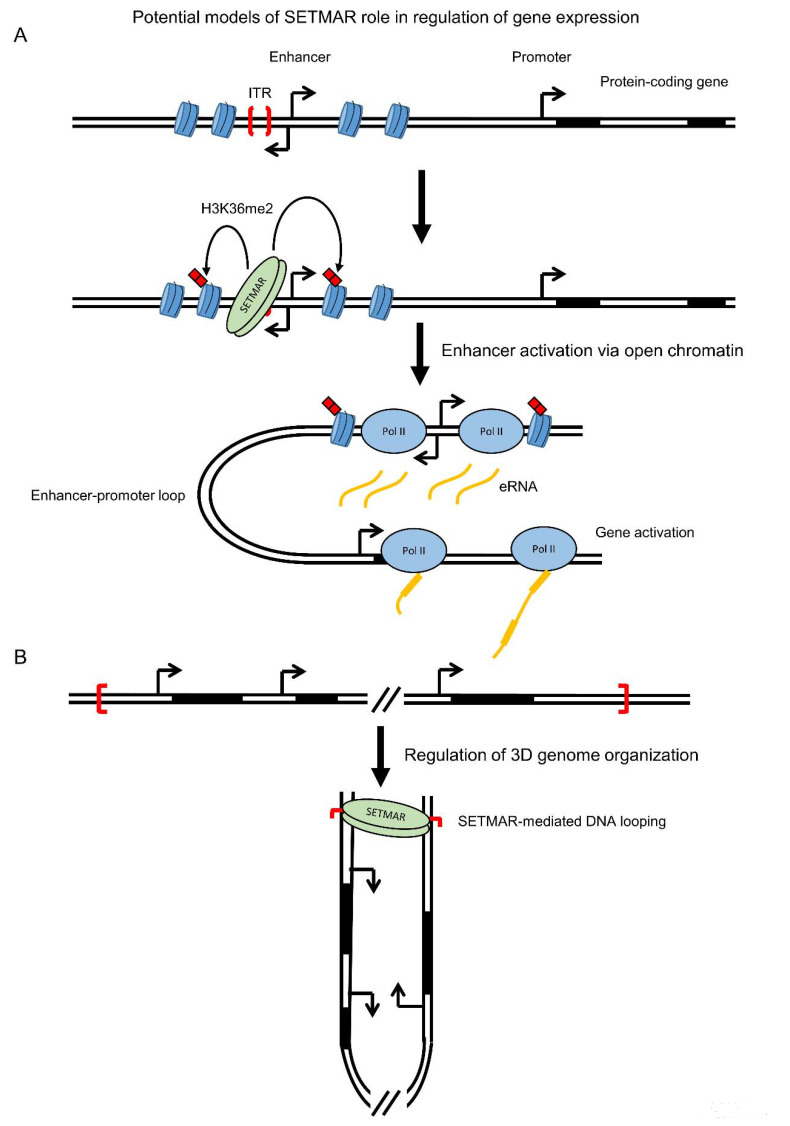
Potential models of SETMAR role in regulation of gene expression. (**A**) SETMAR could regulate gene expression via its binding to enhancer regions. Following SETMAR binding, dimethylation of H3K36 of the neighboring nucleosomes can create an open chromatin environment, via a decrease in H3K27me3, allowing the binding of transcription factors. Activation of the enhancer will then promote transcription of its associated protein-coding gene. (**B**) SETMAR could also regulate gene expression via an involvement in the 3D genome. Binding of SETMAR on two single ITRs can mediate the formation of a DNA loop that could regulate the expression of the genes present in the loop. Additionally, SETMAR could participate in 3D genome organization via the ITRs located near cohesin-CTCF-anchored loops.

**Figure 6 life-11-01342-f006:**
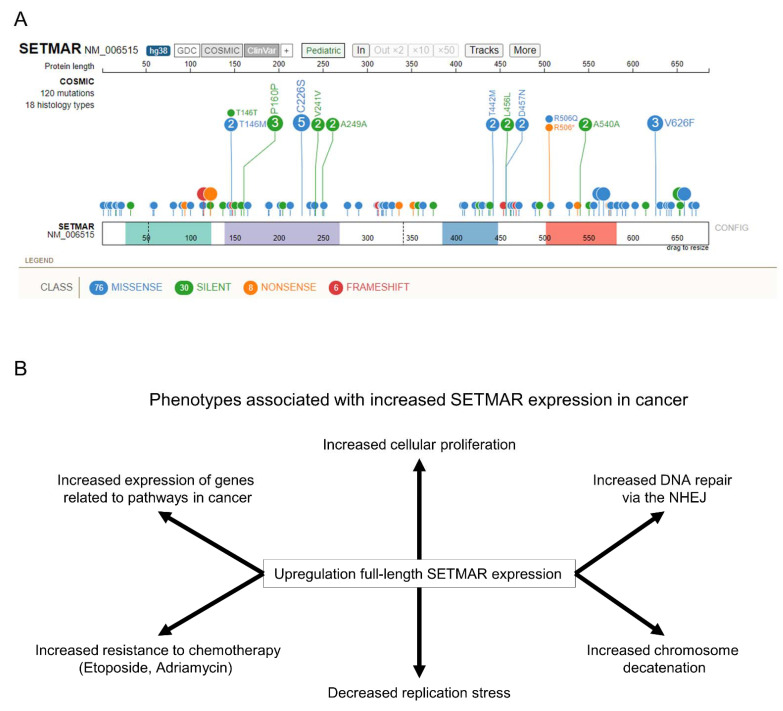
SETMAR roles in disease. (**A**) Visualization of SETMAR known mutations in cancer using ProteinPaint [68]. SETMAR major domains are shown in color at the bottom. The amino acid mutated is shown next to the circle while the number of observations of each mutation is shown within the circle. (**B**) Upregulation of SETMAR expression in cancer is associated with several phenotypes that can promote the tumor’s growth and resistance to treatment.

## Data Availability

Not applicable.

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
