# Peer review of "Structure, Activity, and Function of SETMAR Protein Lysine Methyltransferase"

_life, 2021, doi:10.3390/life11121342_

Round 1
Reviewer 1 Report
The review covers structure and function of an important protein SETMAR which is involved in multiple cellular functions. If author can add some information on the current efforts that are being made towards targeting SETMAR that would be very informative for the readers. Overall, the review is well structured, well written and will be an important contribution to the field.
Author Response
The review covers structure and function of an important protein SETMAR which is involved in multiple cellular functions. If author can add some information on the current efforts that are being made towards targeting SETMAR that would be very informative for the readers. Overall, the review is well structured, well written and will be an important contribution to the field.
Thank you for the positive comments.
The following changes on the current efforts to target SETMAR have been made in the part 5:
“While specific antibodies against SETMAR are available [27,74], a major limitation remains the current lack of tools to study SETMAR in vivo.”
“Current efforts in targeting SETMAR has found ciprofloxacin as an inhibitor of SETMAR transposase domain [70], but the cytotoxicity of this small molecule remains an issue [71-73]. The development of less cytotoxic small molecules targeting SETMAR DNA transposase domain is still needed before potential clinical use. Another important need will be the development of a specific inhibitor of SETMAR methyltransferase activity, which will be critical to determine the targets of SETMAR. In addition, a specific inhibitor could potentially be clinically relevant, by itself or in combination, as SETMAR is dysregulated in several cancers. Another possibility could be the development of small molecules promoting the ubiquitination and degradation of SETMAR by the proteasome, such as PROTACs or molecular glues [77].”
Reviewer 2 Report
SETMAR, a protein (histone) lysine methyltransferase containing an inactive C-terminal DNA transposase domain is unique to anthropoid primates. The well-arranged review by Tellier comprehensively summarises the current knowledge about this remarkable protein, which is involved in DNA double strand break repair and, in addition, seems to play a role in gene regulation.
Minor points and typos:
The author does not mention the co-substrate (methyl donor) used by SETMAR. Is it S-adenosylmethionine?
Line 63: replace “surround” by “flank”.
Line 79: replace “… forms homodimer …” by “… forms a homodimer …”.
Line 87: replace “… isoforms encodes for proteins …” by “… isoforms encode proteins …”.
Line 90: replace “…encoding for full length SETMAR …” by “…encoding full length SETMAR …”.
Line 92: replace “…encodes for a SETMAR protein…” by “…encodes a SETMAR protein…”.
Line 95: replace “regulators” by “regulators”.
Line 109: replace “… by a color” by “… by a different color”.
Lines 117 to 118: replace “… SETMAR have been the most investigated” by “… SETMAR has been the most intensively investigated”.
Line 127: Is H3K115 correct? To my knowledge, H3 consists of 79 amino acids only.
Line 173: replace “regulate” by “regulates”.
Line 180: replace “stabilty” by “stability”.
Line 185: replace “recognized” by “recognize”.
Line 238: replace “… that while each …” by “… that, while each …”
Line 284: replace “paper” by “papers”.
Line 297: replace “promoters” by “promoter”.
Author Response
SETMAR, a protein (histone) lysine methyltransferase containing an inactive C-terminal DNA transposase domain is unique to anthropoid primates. The well-arranged review by Tellier comprehensively summarises the current knowledge about this remarkable protein, which is involved in DNA double strand break repair and, in addition, seems to play a role in gene regulation.
Thank you for the positive comments.
Minor points and typos:
The author does not mention the co-substrate (methyl donor) used by SETMAR. Is it S-adenosylmethionine?
Yes, the following phrase has been added: “The SET domain contains binding sites for the lysine ligand and the co-factor S-adenosylmethionine (SAM), which provides the methyl groups.”
Line 63: replace “surround” by “flank”.
Done.
Line 79: replace “… forms homodimer …” by “… forms a homodimer …”.
Done.
Line 87: replace “… isoforms encodes for proteins …” by “… isoforms encode proteins …”.
Done.
Line 90: replace “…encoding for full length SETMAR …” by “…encoding full length SETMAR …”.
Done.
Line 92: replace “…encodes for a SETMAR protein…” by “…encodes a SETMAR protein…”.
Done.
Line 95: replace “regulators” by “regulators”.
Done.
Line 109: replace “… by a color” by “… by a different color”.
Done.
Lines 117 to 118: replace “… SETMAR have been the most investigated” by “… SETMAR has been the most intensively investigated”.
Done.
Line 127: Is H3K115 correct? To my knowledge, H3 consists of 79 amino acids only.
Yes, the H3K115 residue is however located in the DNA-histone dyad interface, and not in the tail. A new reference has been added (number 46) and the text has been changed to: “Carlson et al found that SETMAR could only weakly methylate histone H3 in vitro, potentially on H3K115, a residue located in the DNA-histone dyad interface [45,46].”
Line 173: replace “regulate” by “regulates”.
Done.
Line 180: replace “stabilty” by “stability”.
Done.
Line 185: replace “recognized” by “recognize”.
Done.
Line 238: replace “… that while each …” by “… that, while each …”
Done.
Line 284: replace “paper” by “papers”.
Done.
Line 297: replace “promoters” by “promoter”.
Done.